# Projective phase measurements in one-dimensional Bose gases

**Yuri D. van Nieuwkerk[1][*], Jörg Schmiedmayer[2] and Fabian H.L. Essler[1]**

**1** Rudolf Peierls Centre for Theoretical Physics, Parks Road, Oxford OX1 3PU
**2** Vienna Center for Quantum Science and Technology (VCQ), Atominstitut,
TU-Wien, Vienna, Austria

[*] yuri.vannieuwkerk@physics.ox.ac.uk

## Abstract

We consider time-of-flight measurements in split one-dimensional Bose gases. It is well known that the low-energy sector of such systems can be described in terms of two compact phase fields $\hat{\phi}_{a,s}(x)$. Building on existing results in the literature we discuss how a single projective measurement of the particle density after trap release is in a certain limit related to the eigenvalues of the vertex operator $e^{i\hat{\phi}_a(x)}$. We emphasize the theoretical assumptions underlying the analysis of "single-shot" interference patterns and show that such measurements give direct access to multi-point correlation functions of $e^{i\hat{\phi}_a(x)}$ in a substantial parameter regime. For experimentally relevant situations, we derive an expression for the measured particle density after trap release in terms of convolutions of the eigenvalues of vertex operators involving both sectors of the two-component Luttinger liquid that describes the low-energy regime of the split condensate. This opens the door to accessing properties of the symmetric sector via an appropriate analysis of existing experimental data.



# 1 Introduction

The purpose of this manuscript is to revisit the theoretical basis for the analysis of matter-wave interferometry experiments on split one-dimensional Bose gases [1–11]. In these experiments a trapped (quasi) one-dimensional Bose gas is first split in two, then allowed to time evolve under an interacting Hamiltonian, released into three-dimensional space and finally measured after a given period of free evolution. The measurement of the particle density after free evolution exhibits interference fringes. Repeating the experimental sequence many times provides an enormous amount of information on the quantum mechanical state of the many-particle system before trap release. Histograms of the observed interference patterns provide the full quantum mechanical distribution function of the measured observable [3–5, 12–17]. The ability of measuring distribution functions of physical observables in interacting many-particle systems (out of equilibrium) is a very exciting feature of cold atom experiments [18], but poses a formidable theoretical problem and so far only few results have been obtained in the literature [13, 19–31]. In the case of split one-dimensional Bose condensates the probability distributions of the observed interference patterns have been analyzed in the framework of Luttinger liquid theory and very good agreement with experimental observations has been found [32–36]. Here we provide a detailed derivation for the fit formula used to analyze the experimental data [5] for individual measurements. The formula is obtained in a particular limit of a new theoretical expression that describes projective density measurements in time of flight experiments. Like previous work our approach is based on the Luttinger liquid description of the phase degrees of freedom. We discuss why this analysis is restricted to the weakly interacting regime, and what modifications emerge for stronger interactions. Our derivation makes it clear why such measurements provide access to equal time multi-point correlation

functions of vertex operators of the phase field.

This paper is organized as follows: in Section 2, we review the setup for time-of-flight experiments and how measured properties are related to quantities in the split gases before trap release. In Section 3, we express the measured density after time-of-flight in terms of an appropriate vertex operator in the field theory describing the low-energy degrees of freedom of the one-dimensional gas. Section 4 shows how to construct a basis of eigenstates for these operators. In Section 5, we show that the experiments can be viewed as projective measurements that sample the eigenvalues of the vertex operator according to a probability distribution that is fixed by the state which the system is initialized in after the splitting procedure.

As an example, we consider the case of coherently split bose gases without tunnel coupling, *cf.* Refs. [35, 36].

## 2 Setup and time-of-flight recombination

We consider a pair of one-dimensional bose gases of length $L$. We denote the longitudinal (along the 1D direction) and transverse coordinates by $x$ and $\vec{r}$ respectively. The corresponding momentum coordinates will be denoted by $(k, \vec{p})$ and we use units such that $\hbar = 1$ throughout the paper. The gases are placed at transverse positions $\vec{r}_{1,2} = \pm \vec{d}/2$. In the first stage of the experiment, the two condensates time-evolve under some one-dimensional Hamiltonian $H_{1d}$, until a time $t_0$. In the second stage, they are released from the trap, causing them to expand in three-dimensional space and overlap. Finally, the three-dimensional gas density is measured after a "time of flight" $t_1$. We model this measurement by assuming that the many-particle wave function collapses to a simultaneous eigenstate $|\Psi\rangle$ of the operators

$$\hat{\rho}_{\text{tof}}(x, \vec{r}, t_1 + t_0) = \hat{\Psi}^\dagger(x, \vec{r}, t_1 + t_0)\hat{\Psi}(x, \vec{r}, t_1 + t_0), \tag{1}$$

where $\hat{\Psi}_{\text{tof}}(x, \vec{r}, t)$ are Heisenberg picture boson annihilation operators at position $(x, \vec{r})$ and time $t$. They satisfy equal-time commutation relations

$$\left[\hat{\Psi}(x, \vec{r}), \hat{\Psi}^\dagger(z, \vec{r}')\right] = \delta(x - z)\delta^2(\vec{r} - \vec{r}'), \tag{2}$$

with all other commutators being zero. Importantly the density operators $\hat{\rho}_{\text{tof}}(x, \vec{r}, t_1 + t_0)$ at different positions commute. This implies that the measurement outcome is the function $\varrho_{\text{tof}}(x, \vec{r}, t_1 + t_0)$ describing the eigenvalues of the density operators on the simultaneous eigenstate $|\Psi\rangle$.

We now turn to the relation between $\hat{\Psi}(x, \vec{r}, t)$ and the field operators $\hat{\psi}_{1,2}(x, t_0)$ describing the two one-dimensional gases at the time $t_0$ of the trap release [37, 38]. We have

$$\hat{\Psi}(x, \vec{r}, t) = U^\dagger(t; t_0)\hat{\Psi}(x, \vec{r}, t_0)U(t; t_0), \tag{3}$$

where $U^\dagger(t; t_0) = T \exp i \int_{t_0}^{t_1} dt H(t)$ is the time evolution operator describing the free expansion after the trap release. This expansion can be analyzed by distinguishing between the "transverse" motion, occurring perpendicular to the one-dimensional gas, and the expansion along the one-dimensional gas direction, which is customarily referred to as "longitudinal". We retain this nomenclature even though we will impose periodic boundary conditions on the one-dimensional gas for simplicity (see Section 3). Open boundary conditions can be accommodated straightforwardly in our approach, but as our focus is on "bulk" physics we leave the discussion of boundary effects to future work. We will make two simplifying assumptions [37, 38] about the expansion of the gas after trap release:

1. The state of the gas before its release factorizes into transverse and longitudinal degrees of freedom. The longitudinal state is the complicated many-body state we are interested in. The transverse degrees of freedom occupy the ground state of a harmonic oscillator potential, with vanishing overlap between the two wells. The wells are assumed to have a large transverse trapping frequency $\omega_\perp$. This implies that the spatial distribution of the transverse state is a spatially narrow Gaussian, ensuring that the velocity distribution in the transverse directions is much broader than in the longitudinal direction. In some works [5,38] it is therefore assumed that the longitudinal degrees of freedom are effectively frozen on the time scales relevant for expansion. Relaxing this simplifying assumption leads to a more involved description [39,40]. In what follows, results based on frozen longitudinal dynamics will be presented alongside results for the full, three-dimensional expansion.

2. The gases are assumed to evolve as free particles after they have been released from the trap. For a justification of this assumption, the reader is referred to [39].

Under these assumptions the time evolution after trap release is described by

$$U(t; t_0) = e^{-i(t-t_0)\left(\hat{P}_x^2 + \hat{\vec{P}}_\perp^2\right)/2m}. \tag{4}$$

Here $\hat{P}_x$ ($\hat{\vec{P}}_\perp$) is the total momentum operator in the longitudinal (transverse) direction and $m$ is the mass of the individual particles. It is now straightforward to obtain the desired relation between the field operators at the time of measurement ($t = t_1 + t_0$) and the time of trap release ($t = t_0$),

$$\hat{\Psi}_{\text{tof}}(x, \vec{r}, t_1 + t_0) = \int \frac{dk \, d^2\vec{p} \, dy \, d^2\tilde{\vec{r}}}{(2\pi)^3} e^{-ik(x-y)} e^{-i\vec{p}\cdot(\vec{r}-\tilde{\vec{r}})} e^{-it_1 \frac{k^2+\vec{p}^2}{2m}} \hat{\Psi}(y, \tilde{\vec{r}}, t_0). \tag{5}$$

From our previous discussion we know that at $t = t_0$ a basis of single-particle states (in the low-energy sector of the Hilbert space) is obtained by having a boson at position $x$ that is the ground state of one of the transverse harmonic oscillators centred at $\pm\vec{d}/2$ in the transverse directions. This implies that the Bose field can be decomposed as

$$\hat{\Psi}(x, \vec{r}, t_0) = \hat{\psi}_1(x, t_0) g(\vec{r} + \vec{d}/2) + \hat{\psi}_2(x, t_0) g(\vec{r} - \vec{d}/2), \tag{6}$$

where $\hat{\psi}_{1,2}(x, t_0)$ creates a boson at position $x$ in the ground state of the transverse harmonic oscillator centred at $\pm\vec{d}/2$ and $g(\vec{r} \pm \vec{d}/2)$ denotes the corresponding ground state wave functions. The Bose fields $\hat{\psi}_i(x, t_0)$ have equal time commutation relations $[\hat{\psi}_i(x, t), \hat{\psi}_j^\dagger(z, t)] = \delta_{i,j}\delta(x - z)$. Inserting the decomposition (6) into (5), using $g(\vec{x}) \sim e^{-\frac{m\omega}{2}\vec{x}^2}$ and assuming that $t_1 \gg 1/\omega$ (where $\omega$ is the frequency of the harmonic potential in the transverse direction) then gives

$$\hat{\Psi}(x, \vec{r}, t_1 + t_0) = f(\vec{r}, t_1) \int dy \, G(x - y, t_1) \left[ \hat{\psi}_1(y, t_0) e^{i\frac{m}{2t_1}(\vec{r}+\vec{d}/2)^2} + \hat{\psi}_2(y, t_0) e^{i\frac{m}{2t_1}(\vec{r}-\vec{d}/2)^2} \right], \tag{7}$$

where the function $f(\vec{r}, t_1)$ is a Gaussian envelope, and $G(x, t_1)$ is a free, single-particle Green's function. The precise form of these functions, together with the details of the calculation, are given in Appendix A.

Using (7) we can identify the observable that is ultimately measured in the time-of-flight experiments as

$$\hat{\rho}_{\text{tof}}(x, \vec{r}, t_1 + t_0) = |f(\vec{r}, t_1)|^2 \iint dy \, dz \, G^*(x-y, t_1) G(x-z, t_1) \left[ \hat{\psi}_1^\dagger(y, t_0)\hat{\psi}_1(z, t_0) \right.$$
$$\left. + \hat{\psi}_2^\dagger(y, t_0)\hat{\psi}_2(z, t_0) + \hat{\psi}_1^\dagger(y, t_0)\hat{\psi}_2(z, t_0)e^{-i\vec{d}\cdot\vec{r}\,m/t_1} + \hat{\psi}_2^\dagger(y, t_0)\hat{\psi}_1(z, t_0)e^{i\vec{d}\cdot\vec{r}\,m/t_1} \right]. \tag{8}$$

Each measurement will select one of the eigenvalues of the above sum of operators. Importantly, the various terms in (8) do not commute with one another. Hence at the level of the "full" Bose gases the measured observable is not simple.

### 2.1 Simplification when the longitudinal expansion is frozen

Denoting by $\hat{\rho}(t_0)$ the density matrix of the system at the time of the trap release, the subsequent evolution is given by

$$\hat{\rho}(t) = U(t; t_0)\hat{\rho}(t_0)U^\dagger(t; t_0) . \tag{9}$$

In cases where $\hat{\rho}(t_0)$ and $t_1$ are such that expansion in the longitudinal direction can be neglected, *cf.* the discussion above, we have

$$\hat{\rho}(t_1 + t_0) \approx \widetilde{U}(t_1 + t_0; t_0)\hat{\rho}(t_0)\widetilde{U}^\dagger(t_1 + t_0; t_0) , \quad \widetilde{U}(t_1 + t_0; t_0) = e^{-it_1\hat{\vec{P}}_\perp^2/2m}. \tag{10}$$

In this case (7) can be replaced by

$$\hat{\Psi}(x, \vec{r}, t_1 + t_0) = f(\vec{r}, t_1)\left[\hat{\psi}_1(x, t_0)e^{i\frac{m}{2t_1}(\vec{r}+\vec{d}/2)^2} + \hat{\psi}_2(x, t_0)e^{i\frac{m}{2t_1}(\vec{r}-\vec{d}/2)^2}\right]. \tag{11}$$

This then results in the following expression for the measured density

$$
\begin{aligned}
\hat{\rho}_{\text{tof}}(x, \vec{r}, t_1 + t_0) &= |f(\vec{r}, t_1)|^2\Big[\hat{\psi}_1^\dagger(x, t_0)\hat{\psi}_1(x, t_0) + \hat{\psi}_2^\dagger(x, t_0)\hat{\psi}_2(x, t_0) \\
&+ \hat{\psi}_1^\dagger(x, t_0)\hat{\psi}_2(x, t_0)e^{-i\vec{d}\cdot\vec{r}\,m/t_1} + \hat{\psi}_2^\dagger(x, t_0)\hat{\psi}_1(x, t_0)e^{i\vec{d}\cdot\vec{r}\,m/t_1}\Big]. 
\end{aligned}
\tag{12}
$$

## 3 Luttinger liquid description of the low-energy degrees of freedom

We have seen how the field operator after time of flight can be related to the separate field operators of the original one-dimensional gases. We focus on the case where the dynamics in the trap is governed by a Hamiltonian of the form

$$H_{\text{1d}} = \sum_{j=1,2}\int_{-L/2}^{L/2}dx\left[\frac{1}{2m}\partial_x\hat{\psi}_j^\dagger(x)\partial_x\hat{\psi}_j(x) + g\,\hat{\psi}_j^\dagger(x)\hat{\psi}_j^\dagger(x)\hat{\psi}_j(x)\hat{\psi}_j(x)\right] + H_{\text{pert}}. \tag{13}$$

We will be interested in cases where $H_{\text{pert}}$ can be considered as a weak perturbation in the sense that it does not change the nature of the low energy degrees of freedom. An example would be a weak tunneling term between the two condensates.

For ease of exposition, we will assume periodic boundary conditions in the one-dimensional bose gas. This means that coordinates $x = \pm L/2$ are associated with each other during evolution under the Hamiltonian (13). After trap release, these points become independent, and the bosons are supported on all of $\mathbb{R}^3$. This somewhat artificial treatment has the advantage that it simplifies our expressions. It must be stressed that a model with open boundary conditions can easily be incorporated into our analysis. Doing so will not, however, change our argument in a fundamental way for regions that are sufficiently far from the edges of the trap.

### 3.1 Low energy projection

In the low-energy sector of the theory dramatic simplifications occur. The low-energy degrees of freedom can be described by bosonization [41]

$$\hat{\psi}_j^\dagger(x) \sim \sqrt{\rho_0 + \frac{\partial_x\hat{\theta}_j(x)}{\pi}}\,e^{-i\hat{\phi}_j(x)}\sum_m A_m e^{2im(\hat{\theta}_j(x)+\pi\rho_0 x)}, \quad j = 1, 2. \tag{14}$$

Here the fields $\partial_x \hat{\theta}_j(x)/\pi$ and $\hat{\phi}_j(x)$ describe long-wavelength fluctuations of density and phase and have commutation relations

$$\left[\frac{\partial_x \hat{\theta}_i(x)}{\pi}, \hat{\phi}_j(z)\right] = i\delta_{i,j}\delta(x-z). \tag{15}$$

The bosonized description applies above a "cutoff" that is set by the healing length $\xi = \pi/mv$ for weakly interacting bosons, with $v$ the velocity of sound. Bosonizing the Hamiltonian (13) leads to a perturbed two-component Luttinger liquid of the form (see Appendix B for details)

$$\mathcal{H} = \sum_{j=s,a} \frac{v}{2\pi} \int_{-L/2}^{L/2} dx \left[K(\partial_x\hat{\phi}_j(x))^2 + \frac{1}{K}(\partial_x\hat{\theta}_j(x))^2\right] + \mathcal{H}_{\text{pert}}, \tag{16}$$

where $\mathcal{H}_{\text{pert}}$ is the low-energy projection of $H_{\text{pert}}$ and where we have defined symmetric and antisymmetric combinations of the fields by

$$\hat{\phi}_a = \hat{\phi}_1 - \hat{\phi}_2, \ \ \hat{\phi}_s = \hat{\phi}_1 + \hat{\phi}_2, \ \ \hat{\theta}_a = \frac{\hat{\theta}_1 - \hat{\theta}_2}{2}, \ \ \hat{\theta}_s = \frac{\hat{\theta}_1 + \hat{\theta}_2}{2}. \tag{17}$$

In order for (16) to apply we require that $\mathcal{H}_{\text{pert}}$ can be treated as a perturbation in the sense that it does not invalidate a low-energy description in terms of phase fields. An example [17, 42] is a small tunneling term (with $\lambda \ll mv^2$ proportional to the tunneling amplitude)

$$H_{\text{pert}} = \lambda \int dx \left[\hat{\psi}_1^\dagger(x)\hat{\psi}_2(x) + \text{h.c.}\right], \tag{18}$$

giving a relevant (in the renormalization group sense) perturbation of the form

$$\mathcal{H}_{\text{pert}} = \lambda' \int dx \ \cos\hat{\phi}_a(x). \tag{19}$$

## 3.2 Case with no longitudinal expansion and weak interactions

We first discuss the simpler case in which the longitudinal expansion is assumed to be negligible. Applying the bosonization identity (14) to the observable measured in time-of-flight experiments, the measured density operator (12) takes the form

$$\hat{\rho}_{\text{tof}}(x, \vec{r}, t_1 + t_0) \simeq 2\left|f(\vec{r}, t_1)\right|^2 \left\{ |A_0|^2\left(\rho_0 + \Pi_s(x, t_0)\right)\left(1 + \text{Re}\left[e^{i\hat{\phi}_a(x,t_0) + i\vec{d}\cdot\vec{r}\frac{m}{t_1}}\right]\right)\right.$$

$$+ 4A_0 A_1\left[\left(\rho_0 + \Pi_s(x, t_0)\right)\cos\left(2\hat{\theta}_s(x,t_0) + 2k_F x\right)\cos 2\hat{\theta}_a(x,t_0)\left[1 + \text{Re}\left(e^{i\hat{\phi}_a(x,t_0) + i\vec{d}\cdot\vec{r}\frac{m}{t_1}}\right)\right]\right.$$

$$\left.\left. - \Pi_a(x,t_0)\sin\left(2\hat{\theta}_s(x,t_0) + 2k_F x\right)\sin 2\hat{\theta}_a(x,t_0)\right] + \dots\right\}, \tag{20}$$

where we have defined

$$\Pi_\alpha(x,t_0) = \frac{\partial_x \hat{\theta}_\alpha(x,t_0)}{\pi}, \quad \alpha = a,s. \tag{21}$$

Here the dots refer to subleading terms in the expansion, in the sense that the operators have higher scaling dimensions. These operators can have nonzero expectation values on the states of interest, and they are multiplied by coefficients $A_{m\neq 0}$. In fact, it has been shown [43] that if $K$ is close to 1, $A_0$ and $A_1$ approach each other, and higher order terms cannot simply be neglected.

The weakly interacting regime $K \gg 1$ is of particular interest in view of existing experiments. Here the coefficients $A_{m \neq 0}$ are small and we need to retain only the first line of (20), if the longitudinal expansion during time-of-flight is neglected. This gives

$$\hat{\rho}_{\text{tof}}(x, \vec{r}, t_1 + t_0)\Big|_{K \gg 1} \simeq 2|A_0|^2 |f(\vec{r}, t_1)|^2 \left(\rho_0 + \frac{\partial_x \hat{\theta}_s(x, t_0)}{\pi}\right)\left(1 + \text{Re}\left[e^{i\hat{\phi}_a(x, t_0) + i\vec{d} \cdot \vec{r} m/t_1}\right]\right). \quad (22)$$

As $[\partial_x \hat{\theta}_s(x, t_0), e^{i\hat{\phi}_a(x, t_0)}] = 0$, a projective measurement of $\hat{\rho}_{\text{tof}}$ projects onto simultaneous eigenstates of these operators.

### 3.2.1 Relation of operator eigenvalues to experimental fit formulas

In (22) the measured density operator has been expressed as a function of commuting operators $e^{i\hat{\phi}_a(x)}$. A measurement then projects onto a simultaneous eigenstate of these operators. Let us denote the corresponding eigenvalues by the functions $e^{i\varphi_a(x)}$. In the case at hand, i.e. negligible longitudinal expansion, the density measurement then returns the eigenvalue

$$\varrho_{\text{tof}}(x, \vec{r}, t_1 + t_0) \approx 2\rho_0 |A_0|^2 |f(\vec{r}, t_1)|^2 \left(1 + \text{Re}\left[e^{i\varphi_a(x, t_0) + i\vec{d} \cdot \vec{r} m/t_1}\right]\right), \quad (23)$$

where it has been assumed that the relevant eigenvalues of $\partial_x \hat{\theta}_s$ are much smaller than $\rho_0$. This assumption is justified if the symmetric sector is in a thermal state [36], where density fluctuations are small [44].

In many experiments [4, 5] the measured gas density is integrated over a distance $l$ along the longitudinal coordinate of the gas, giving the measured eigenvalue

$$
\begin{aligned}
R_{\text{tof}}(\vec{r}, t_1 + t_0, \ell) &= \int_{-\ell/2}^{\ell/2} dx \, \varrho_{\text{tof}}(x, \vec{r}, t_1 + t_0) \\
&\approx 2\rho_0 |A_0|^2 |f(\vec{r}, t_1)|^2 \left(\ell + \text{Re}\left[e^{i\vec{d} \cdot \vec{r} m/t_1} \int_{-\ell/2}^{\ell/2} dx \, e^{i\varphi_a(x, t_0)}\right]\right). \quad (24)
\end{aligned}
$$

This can now be directly compared to the formula used to fit the experimentally measured interference fringes given in [5] as

$$\tilde{R}_{\text{tof}}(\vec{r}, t_1 + t_0, \ell) = 2\rho_0 \ell |A_0|^2 |f(\vec{r}, t_1)|^2 \left(1 + C(\ell, t_0) \cos\left(\Phi(\ell, t_0) + \vec{d} \cdot \vec{r} m/t_1\right)\right). \quad (25)$$

Comparing (25) and (24) shows that the quantities $C(\ell)$ and $\Phi(\ell)$ are related to the measured eigenvalues $e^{i\varphi_a(x)}$ by

$$C(\ell, t_0) e^{i\Phi(\ell, t_0)} = \frac{1}{\ell} \int_{-\ell/2}^{\ell/2} dx \, e^{i\varphi_a(x, t_0)}. \quad (26)$$

### 3.2.2 Determining multipoint correlation functions from measurements

The previous discussion has shown that the experimental measurement of individual interference patterns permits the determination of the corresponding vertex-operator eigenvalues $e^{i\varphi_a(x)}$. Having these in hand it is then possible to extract (connected) multi-point correlation functions from the measurements as follows [17, 45]. Expectation values of the form

$$g_{\alpha_1, \dots, \alpha_n}(x_1, x_2, \dots, x_n) \equiv \langle \psi(t) | \prod_n e^{i\alpha_n \hat{\phi}(x_n)} | \psi(t) \rangle \quad (27)$$

are obtained by averaging over many measurements of "single-shot" interference patterns. According to our previous discussion, each such measurement provides the eigenvalue $e^{i\varphi_a(x)}$

of $e^{i\hat{\phi}_a(x)}$. As vertex operators at different positions commute with one another, their respective measurements are independent. Hence the outcome for measuring only $\prod_n e^{i\alpha_n\hat{\phi}(x_n)}$ is simply given by the product of the corresponding eigenvalues $\prod_n e^{i\alpha_n\varphi_a(x_n)}$. These are straightforwardly extracted from the single-shot measurements discussed above by considering fixed positions $x_1, \ldots, x_n$. Averaging over the outcomes of a large number of such measurements, and keeping the positions $x_1, \ldots, x_n$ fixed throughout provides the desired expectation values (27).

## 3.3 General case in the weakly interacting regime

We now turn to the case where the longitudinal expansion is not negligible. In order to have manageable expressions we constrain our discussion to the regime of weak interactions $K \gg 1$, where we can set the amplitudes $A_{n\geq 1} = 0$. Applying the bosonization identity (14) we then find

$$\hat{\rho}_{\text{tof}}(x, \vec{r}, t_1 + t_0) \simeq 2|f(\vec{r}, t_1)|^2 |A_0|^2 \iint dy\, dz\, G^*(x-y, t_1)G(x-z, t_1)$$

$$\times \left\{ \left( \rho_0 + \frac{\partial_y \hat{\theta}_1(y, t_0) + \partial_z \hat{\theta}_1(z, t_0)}{2\pi} \right) e^{-i(\hat{\phi}_1(y,t_0) - \hat{\phi}_1(z,t_0))} + (1 \to 2) \right. \tag{28}$$

$$\left. + \left[ \left( \rho_0 + \frac{\partial_y \hat{\theta}_1(y, t_0) + \partial_z \hat{\theta}_2(z, t_0)}{2\pi} \right) e^{-i(\hat{\phi}_1(y,t_0) - \hat{\phi}_2(z,t_0))} \right] e^{-i\vec{d}\cdot\vec{r}\, m/t_1} + [1 \leftrightarrow 2]\, e^{i\vec{d}\cdot\vec{r}\, m/t_1} \right\} + \ldots.$$

This expression involves products of non-commuting operators, which we must diagonalize in order to develop a theory of projective measurements. This significant complication vanishes in the experimentally relevant case when density fluctuations are small compared to the average density $\rho_0$ [44]. In that case, the fields $\partial_x \hat{\theta}_{1,2}$ may be neglected, so that the measured density operator becomes

$$\hat{\rho}_{\text{tof}}(x, \vec{r}, t_1 + t_0)\Big|_{K \gg 1} \simeq \rho_0 \Big| A_0 f(\vec{r}, t_1) \int dy\, G(x-y, t_1) \Big[ e^{i\frac{m}{2t_1}\vec{r}\cdot\vec{d}} e^{\frac{i}{2}(\hat{\phi}_s(y,t_0) + \hat{\phi}_a(y,t_0))}$$

$$+ e^{-i\frac{m}{2t_1}\vec{r}\cdot\vec{d}} e^{\frac{i}{2}(\hat{\phi}_s(y,t_0) - \hat{\phi}_a(y,t_0))} \Big] \Big|^2. \tag{29}$$

This expression only contains fields which mutually commute. A measurement thus projects onto simultaneous eigenstates of these fields, based on some probability distribution which is set by the state at the time of release. A projective measurement returns the eigenvalues

$$\varrho_{\text{tof}}(x, \vec{r}, t_1 + t_0) \simeq \rho_0 \Big| A_0 f(\vec{r}, t_1) \int dy\, G(x-y, t_1) \Big[ e^{i\frac{m}{2t_1}\vec{r}\cdot\vec{d}} e^{\frac{i}{2}(\varphi_s(y,t_0) + \varphi_a(y,t_0))}$$

$$+ e^{-i\frac{m}{2t_1}\vec{r}\cdot\vec{d}} e^{\frac{i}{2}(\varphi_s(y,t_0) - \varphi_a(y,t_0))} \Big] \Big|^2, \tag{30}$$

where $e^{i\varphi_{a,s}(x,t_0)}$ are the corresponding eigenvalues of $e^{i\hat{\phi}_{a,s}(x,t_0)}$.

## 4 Vertex operator eigenstates

We now turn to the construction of eigenstates of the vertex operators $e^{i\hat{\phi}_a(x)}$ and corresponding eigenvalues $e^{i\varphi_a(x)}$. The mode expansions for $\hat{\phi}_a(x)$ and $\partial_x \hat{\theta}_a(x)$ are given in Appendix B and involve zero modes that reflect the compact nature of the phase fields $\hat{\phi}_a(x)$. In particular we have $\hat{\phi}_a(x+L) = \hat{\phi}_a(x) + 2\pi\hat{J}_a$, where the eigenvalues $j_a$ of $\hat{J}_a$ are integers. We will

consider cases in which the dynamics occurs in the $j_a = 0$ subspace, i.e. the initial states lie in this subspace and $[\hat{J}_a, \mathscr{H}] = 0$. This leaves us with mode expansions of the form

$$\hat{\phi}_a(x) = \sum_j u_j \left( \hat{a}_j - \hat{a}_{-j}^{\dagger} \right) e^{iq_j x}, \tag{31}$$

$$\frac{\partial_x \hat{\theta}_a(x)}{\pi} = \frac{-i}{2u_0 L} \left( \hat{a}_0 + \hat{a}_0^{\dagger} \right) + \sum_{j \neq 0} \frac{i}{2u_j L} \left( \hat{a}_j + \hat{a}_{-j}^{\dagger} \right) e^{iq_j x}, \tag{32}$$

where $q_j = 2\pi j/L$, $\left[ \hat{a}_j, \hat{a}_k^{\dagger} \right] = \delta_{j,k}$ and

$$u_j = \begin{cases} \left| \frac{\pi}{2q_j LK} \right|^{1/2} \mathrm{sgn}\left( q_j \right), & \text{for } j \neq 0, \\ \frac{i}{4} \sqrt{\frac{2v}{K}} & \text{for } j = 0. \end{cases} \tag{33}$$

As $[\hat{a}_k - \hat{a}_{-k}^{\dagger}, \hat{a}_n - \hat{a}_{-n}^{\dagger}] = 0$ the eigenvalue equation $e^{i\hat{\phi}_a(x)} |\{f_n\}\rangle = e^{i\varphi_a(x)} |\{f_n\}\rangle$ then separates into equations for the individual modes

$$u_k \left( \hat{a}_k - \hat{a}_{-k}^{\dagger} \right) |\{f_n\}\rangle = f_k |\{f_n\}\rangle. \tag{34}$$

Here the eigenvalues $f_k$ are the Fourier coefficients of the function $\varphi_a(x)$

$$\varphi_a(x) = \sum_{j=0}^{\infty} f_j \, e^{iq_j x}. \tag{35}$$

As $\hat{\phi}_a(x)$ is a real field we have $f_{-n}^* = f_n$ and $f_0^* = f_0$. The solution of (34) is

$$|\{f_n\}\rangle_a = \mathscr{N}_f \exp \sum_k \left( \frac{1}{2} \hat{a}_k^{\dagger} \hat{a}_{-k}^{\dagger} + \frac{f_k}{u_k} \hat{a}_k^{\dagger} \right) |0\rangle_a, \tag{36}$$

where $\hat{a}_k |0\rangle_a = 0$. The normalization constant is

$$\mathscr{N}_f = \left( \frac{1}{2\pi |u_0|^2} \right)^{1/4} e^{-\frac{1}{4|u_0|^2} f_0^2} \prod_{k>0} \left( \frac{1}{\pi |u_k|^2} \right)^{1/2} e^{-\frac{1}{2|u_k|^2} |f_k|^2} \tag{37}$$

and ensures the normalization of the eigenstates to delta-functions (see Appendix C for details)

$$\langle \{g_n\} | \{f_n\} \rangle_a = \delta(g_0 - f_0) \prod_{k>0} \delta\left( \mathrm{Re}(g_k - f_k) \right) \delta\left( \mathrm{Im}(g_k - f_k) \right). \tag{38}$$

## 5 Application to coherently split Bose gases

We now specialize to the case of coherently split Bose gases in the absence of tunnel coupling. This setup has been extensively studied in the literature, see e.g. [35,36]. The low-energy limit of this problem is particularly simple, because the symmetric and antisymmetric sectors decouple, and the relevant dynamics occurs only in the latter. The Hamiltonian in the antisymmetric sector is

$$\mathscr{H}_a = \frac{\pi v (\delta \hat{N})^2}{2KL} + \sum_{q \neq 0} v|q| \hat{a}_q^{\dagger} \hat{a}_q. \tag{39}$$

As we are dealing with a free theory the initial state is fixed by specifying the two-point function after the splitting process. In Refs [35, 36] this was taken to be of the form

$$\left\langle \frac{\partial_x \hat{\theta}(x)}{\pi} \frac{\partial_y \hat{\theta}(y)}{\pi} \right\rangle_a = \frac{\rho}{2} \delta_\xi(x-y), \tag{40}$$

where $\delta_\xi(x-y)$ is a delta function which is smeared over the healing length $\xi$. The corresponding state is

$$|W\rangle_a = \mathcal{N}_W \exp\left( \frac{1}{2} \sum_{k\neq 0} W_k \hat{a}_k^\dagger \hat{a}_{-k}^\dagger \right) |0\rangle_a |\psi_{k=0}\rangle, \tag{41}$$

where

$$W_k = \frac{1-\alpha_k}{1+\alpha_k}, \qquad \alpha_k = \frac{|k|K}{\pi\rho}, \quad \langle n|\psi_{k=0}\rangle_a = \left(\frac{1}{\pi\rho_0 L}\right)^{1/4} \exp\left(-\frac{1}{2\rho_0 L}n^2\right), \tag{42}$$

with $|n\rangle$ the eigenstate of $\delta N_a$ with eigenvalue $n$. To connect as closely as possible to the existing literature we adopt the choice (41) in what follows but note that our analysis can be straightforwardly adapted to other initial states.

The Hamiltonian in the symmetric sector is of precisely the same form as (39). For simplicity we will assume the symmetric sector to start out in a Fock state

$$|\psi\rangle_s = |\{n_q\}\rangle, \tag{43}$$

with occupation numbers that follow a Bose-Einstein distribution

$$n_k = \frac{1}{e^{\beta\nu|k|}-1}. \tag{44}$$

Initializing the symmetric sector in a thermal state is common in the literature [36] and rests upon the assumption that the symmetric sector is not affected by the splitting procedure, so that it inherits the thermal properties of the gas before splitting. Since (43) is an eigenstate of the symmetric sector Hamiltonian, and mixing between sectors does not occur, the symmetric sector will be in the state (43) for all times.

In this Section, we will first express the initial state $|W\rangle_a$ in terms of the eigenstates $|\{f_n\}\rangle_a$ of the vertex operator. Using the simple harmonic oscillator form of the Hamiltonian (39), we will then describe time evolution of the overlap coefficients, and interpret these as a probability distribution for the eigenvalues of $\hat{\rho}_{\text{tof}}(x, \vec{r}, t)$, which are directly measured in experiment.

## 5.1 Overlap coefficients

### 5.1.1 Antisymmetric sector

The overlap coefficients $\langle\{f_n\}|W(t)\rangle_a$ can be represented as products over the modes. The contributions from the finite momentum modes are obtained in complete analogy to Appendix C. The zero modes require a separate consideration, which is given in Appendix D. Combining the two kinds of contributions gives the result

$$\left|\langle\{f_n\}|W(t_0)\rangle_a\right|^2 = \sqrt{2\pi c_0(t_0)} \prod_{k\geq 0} \frac{1}{2\pi c_k(t_0)} \exp\left(-\frac{(\mathrm{Re}f_k)^2 + (\mathrm{Im}f_k)^2}{2c_k(t_0)}\right), \tag{45}$$

where we have defined the time-dependent variances

$$c_k(t_0) = \begin{cases} \frac{1}{4\rho_0 L}\left(\cos^2(v|k|t_0) + \left(\frac{k_c}{k}\right)^2 \sin^2(v|k|t_0)\right) & \text{if } k \neq 0, \\ \frac{1}{2\rho_0 L}\left(1 + (vk_c t_0)^2\right) & \text{if } k = 0. \end{cases} \tag{46}$$

The momentum scale occurring here is given by $k_c = 2\pi/\xi$, where $\xi$ is the healing length of the gas. Any fluctuations below this length scale are not captured by the low-energy effective Luttinger Liquid theory.

### 5.1.2 Symmetric sector

To describe the effects of longitudinal expansion, operators in the symmetric sector must be included in the density operator, via (29). In analogy with the antisymmetric sector, a measurement then corresponds to a projection to simultaneous eigenstates $|\{f_q\}\rangle_s$ of $e^{i\hat{\phi}_s(x)}$ in the symmetric sector. These eigenstates will have the same form as their antisymmetric counterparts, presented in (36). The probability of measuring the corresponding eigenvalue $e^{i\varphi_s(x)}$ will similarly be given by the squared overlap with the state of the system in the symmetric sector.

Assuming the symmetric sector to occupy the state (43) at all times, the overlap coefficients with the eigenstates $|\{f_q\}\rangle_s$ of $e^{i\hat{\phi}_s(x)}$ are computed in Appendix E, and read

$$\left|\langle\{f_q\}|\psi\rangle_s\right|^2 = \prod_{q>0} \frac{1}{\pi|u_q|^2} L_{n_q}^2\left(\left|\frac{f_q}{u_q}\right|^2\right) e^{-\left|\frac{f_q}{u_q}\right|^2}, \tag{47}$$

where $L_n(x)$ is the Laguerre polynomial of degree $n$.

## 5.2 Analysis of vertex operator eigenvalue distributions

The squared overlap coefficients (45) have a clear physical interpretation: when measuring $\hat{\rho}_{\text{tof}}(x, \vec{r}, t_1 + t_0)$, the overlap coefficient $\left|\langle\{f_n\}|W(t_0)\rangle_a\right|^2$ gives the probability of collapsing to a state for which $e^{i\hat{\phi}_a(x,t_0)}$ has eigenvalue $e^{i\varphi_a(x,t_0)}$, with

$$\varphi_a(x, t_0) = \sum_j f_j e^{ip_j x}. \tag{48}$$

Examples of typical configurations $\varphi_a(x, t_0)$ are shown in Fig. 1. We first consider the situation at $t_0 = 0$. In that case the coefficients $f_j$ are drawn from a Gaussian distribution with mean 0 and variance $c_j(t_0) = 1/(4\rho_0 L)$. This results in a $\varphi_a(x)$ with vanishing average and short-wavelength variations of size $K^{-1/2}$ as shown in the left panel of Fig. 1. For $t > 0$ the eigenvalues $\varphi_a(x)$ have the structure shown in right panel of Fig. 1. At short wavelengths the variations remain small, while the long wavelength variations become large. The cross-over scale between the two behaviours has been determined by Kitagawa et al. [36], and is given by $l_0 = 8K^2/\rho\pi^2$. It is indicated by a green bar in the right panel of Fig. 1.

## 5.3 Experimental parameters

In order to facilitate a comparison with experimental data, we use the following parameters from [5] in all plots: after splitting, each of the two gases has one-dimensional density $\rho_0 = 45\,\mu\text{m}^{-1}$, healing length $\xi = \hbar\pi/mv = \pi \times 0.42\,\mu\text{m}$ and longitudinal size $L = 80\,\xi$. When applied to Rubidium atoms, this translates to $L \approx 106\,\mu\text{m}$, with a sound velocity given by $v \approx 1.738 \cdot 10^{-3}\,\text{m/s}$. The symmetric sector is in a thermal state, for which we choose $k_{\text{B}}T$

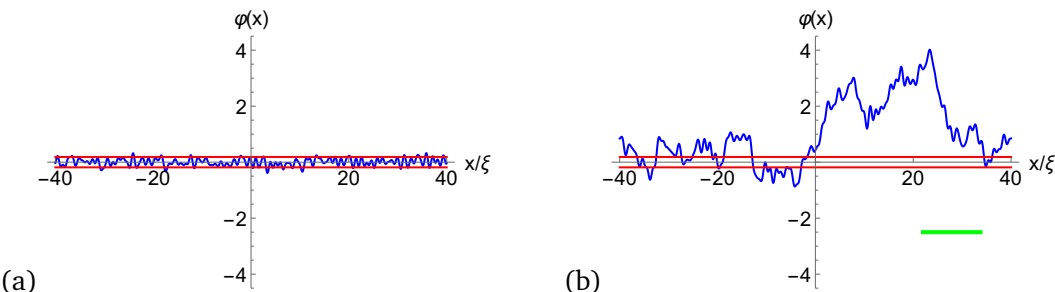

Figure 1: Individual realizations of the eigenvalue $\varphi_a(x, t_0)$ for the phase field $\hat{\phi}_a(x, t_0)$. The typical behavior at $t_0 = 0\,\xi/v$, cf. (a), is distinctly different from that at $t_0 = 14\,\xi/v$, cf. (b). At $t_0 = 0$, small fluctuations occur at all length scales, with a typical amplitude given by $1/\sqrt{K}$ (red lines). At later times, the typical fluctuations are larger for longer length scales. The crossover length scale from which fluctuations become large is indicated with a green bar. In terms of Luttinger Liquid parameters, it is predicted [36] to be $l_0 = 8K^2/\rho\pi^2$. A further note about experimental parameters is presented in Section 5.3.

to be some fraction of $\hbar\omega_\perp$, with transverse trapping frequency $\hbar\omega_\perp = 2\pi \times 1.4\,\text{kHz}$. The state (41) of the antisymmetric sector is not thermal, but it has an energy density given by $\pi v/(3\xi^2)$. To compare this to the energy scale of the symmetric sector, we note that a thermal state with the same energy would be at a temperature of approximately $14\,\text{nK}$, for the parameters presented here. In all plots, the separation between the split gases is taken to be $d = 3\,\mu\text{m}$.

## 6 Results for density measurements after expansion

We now return to the (approximate) expression for the gas density after time of flight, given by (22),

$$\hat{\rho}_{\text{tof}}(x, \vec{r}, t_1 + t_0) \cong 2\rho_0|A_0|^2 |f(\vec{r}, t_1)|^2 \left( 1 + \text{Re}\left[ e^{i\hat{\phi}_a(x, t_0) + i\vec{d}\cdot\vec{r}\,m/t_1} \right] \right), \tag{49}$$

which is valid when longitudinal expansion can be neglected (in the general case one instead uses (29)). A measurement causes the system to collapse to an eigenstate of this operator and concomitantly a simultaneous eigenstate of $e^{i\hat{\phi}_a(x, t_0)}$. The measurement outcome corresponds to the eigenvalues

$$\varrho_{\text{tof}}(x, \vec{r}, t_1 + t_0) \cong 2\rho_0|A_0|^2 |f(\vec{r}, t_1)|^2 \left( 1 + \text{Re}\left[ e^{i\varphi_a(x, t_0) + i\vec{d}\cdot\vec{r}\,m/t_1} \right] \right), \tag{50}$$

where $\varphi_a(x, t_0)$ is characterized by its Fourier coefficients $f_k$. The probability to measure an eigenvalue $\varrho_{\text{tof}}(x, \vec{r}, t_1 + t_0)$ with a corresponding set of Fourier coefficients $\{f_k\}$ is given by the overlap coefficient with the state of the system at the time of release. These overlap coefficients can be computed in specific cases, as we have demonstrated for the case of coherently split bose gases without tunnel-coupling, presented in (45). A completely analogous procedure can be used to describe a measurement of the observable in eqn (29), which requires additional overlaps in the symmetric sector, such as those presented in (47).

With the above formalism in place, experiments can then be modelled as follows. We assume that our system is initialized in the state

$$|\Psi(0)\rangle = |W\rangle_a \otimes |\psi\rangle_s \,, \tag{51}$$

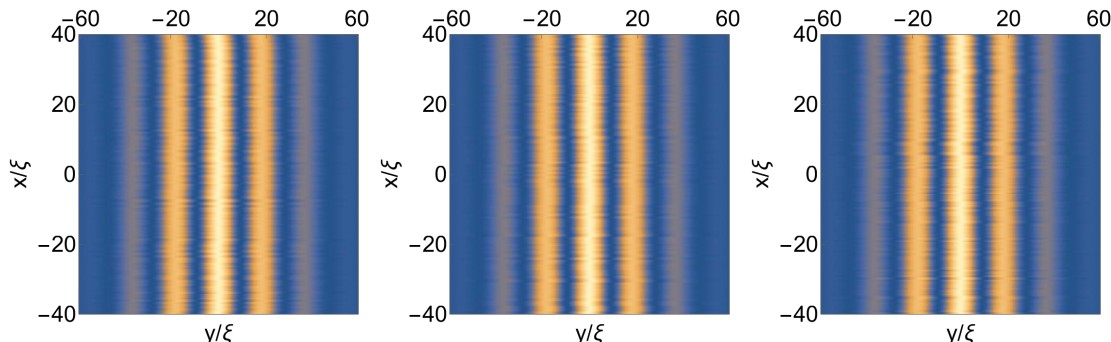

Figure 2: Samples of outcomes for individual (simultaneous) measurements of $\hat{\rho}_{\text{tof}}(x, \vec{r}, t_1 + t_0)$ at $t_0 = 0$, using (49). In all density plots, outcomes are displayed at transverse coordinate $\vec{r} = (y, 0)$. The parameters are as presented in Section 5.3 and the time of flight is taken as $t_1 = 16\,\text{ms}$.

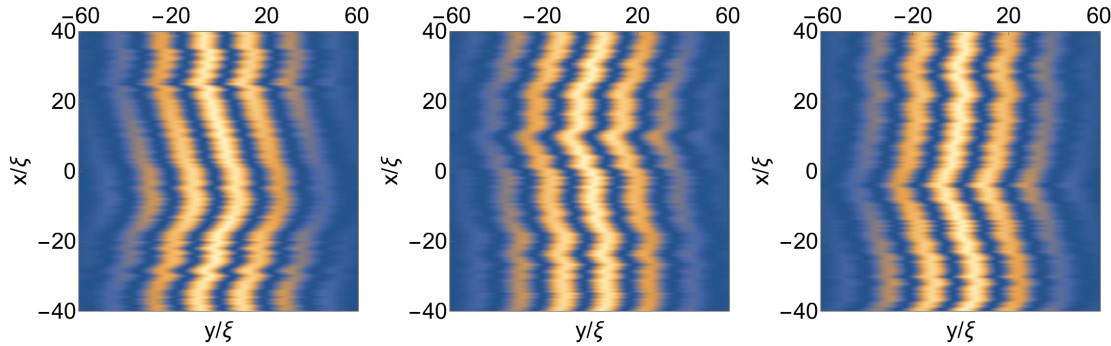

Figure 3: Samples of outcomes for individual (simultaneous) measurements of $\hat{\rho}_{\text{tof}}(x, \vec{r}, t_1 + t_0)$ at $t_0 = 14\,\xi/v \approx 10.6\,\text{ms}$, using (49). The parameters are as presented in Section 5.3 and the time of flight is taken as $t_1 = 16\,\text{ms}$.

where $|W\rangle_a$ and $|\psi\rangle_s$ are given in (41) and (43) respectively. We then let the system evolve under the Luttinger liquid Hamiltonian (16) for a time $t_0$. At time $t_0$ we switch the time evolution to a free expansion and perform a projective density measurement at time $t_0 + t_1$. Some representative results for $\rho_{\text{tof}}(x, \vec{r}, t_1 + t_0)$ evaluated using the simplified expression (50) are presented in Figs 2 and 3. Here the time of flight is taken to be $t_1 = 16\,\text{ms}$.

We see that after a sufficiently long time of flight the measured density exhibits a number of "interference fringes" in the transverse direction. In the initial state ($t_0 = 0$) these are straight, but if the split condensate is left to time evolve ($t_0 > 0$) they start bending. We stress that the intensity along a given fringe does not vary with $x$. This is a property of the simplified expression (50) which assumes that the longitudinal expansion and the density fluctuations in the symmetric sector are negligible. Retaining the term proportional to $\partial_x \hat{\theta}_s$ in (22) does introduce variations in the intensity of the individual fringes. Examples of such realizations are presented in Fig. 4.

## 6.1 Effects of the longitudinal expansion

When the effects of longitudinal expansion are included via (29) the measured density operator is no longer exclusively a function of the relative phase operator but now includes the phase operator from the symmetric sector as well. This dependence on $e^{i\hat{\phi}_s(x)}$ is modeled in complete analogy to our discussion of $e^{i\hat{\phi}_a(x)}$: we construct its eigenstates, compute their squared overlap with the state of the system (51), and interpret this as a probability distribution for

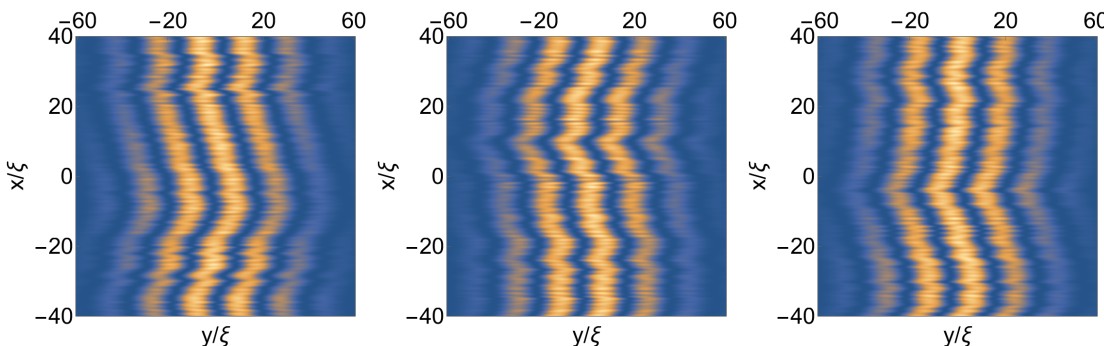

Figure 4: Samples of outcomes for individual (simultaneous) measurements of $\hat{\rho}_{\text{tof}}(x, \vec{r}, t_1 + t_0)$ at $t_0 = 14\,\xi/v$ and $t_1 = 16\,\text{ms}$. The plots were produced using (22), including the term proportional to $\partial_x \hat{\theta}_s$. The temperature in the symmetric sector is $34\,\text{nK}$, which corresponds to $k_B T = 0.5\,\hbar\omega_\perp$, using the parameters presented in Section 5.3.

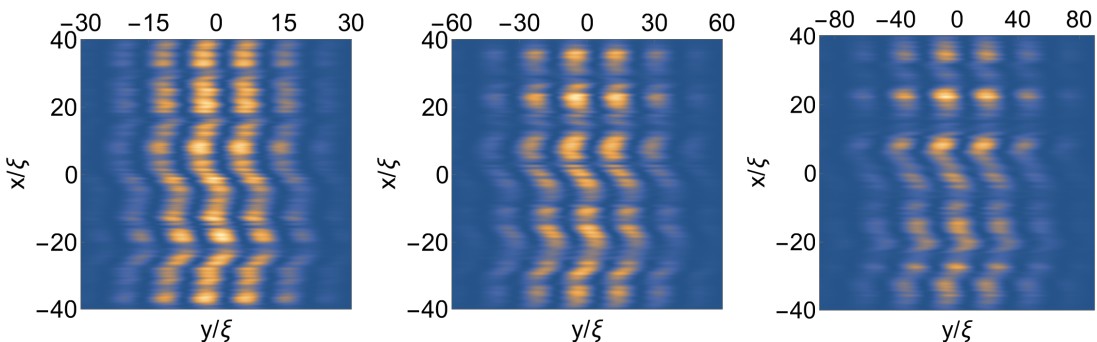

Figure 5: Outcomes for a single projective measurement of $\hat{\rho}_{\text{tof}}(x, \vec{r}, t_1 + t_0)$, using (29), observed for different time-of-flight values $t_1$ and at fixed one-dimensional evolution time $t_0 = 14\,\xi/v$. The temperature in the symmetric sector is $34\,\text{nK}$, which corresponds to $k_B T = 0.5\,\hbar\omega_\perp$, using the parameters from Section 5.3. From left to right, the time of flight is $t_1 = 8, 16$ and $24\,\text{ms}$, respectively. The underlying eigenvalues $e^{i\varphi_{a,s}(x,t_0)}$ are taken to be identical in all three plots in order to accentuate the effects of the time of flight.

the corresponding eigenvalues.

A comparison between this improved analysis (which employs the overlaps computed in Section 5.1) and the case of frozen longitudinal dynamics is presented in Fig. 5. It can be observed that additional "density ripples" emerge in the longitudinal direction, as a consequence of interference between points with different longitudinal coordinates in the original two gases. These density ripples become more pronounced as the time of flight $t_1$ increases, and they occur on longer length scales: whereas $\varrho_{\text{tof}}(x, \vec{r}, t_0 + t_1)$ only involves operators at position $x$ at $t_1 = 0$, it acquires contributions from points at an increasingly large longitudinal separation as $t_1$ increases. This effect is sensitive to the temperature in the symmetric sector, as is illustrated in Fig. 6. A detailed analysis of these density ripples in the density-density correlation function, including their temperature dependence, has been presented in [39, 40].

### 6.1.1 On extracting the eigenvalues $e^{i\varphi_a(x,t_0)}$ from $\varrho_{\text{tof}}(x, \vec{r}, t_1 + t_0)$

Although the effects of longitudinal expansion included in (30) ensure a realistic description of the observed gas density, they complicate the extraction of the eigenvalues $e^{i\varphi_a(x,t_0)}$, due to the

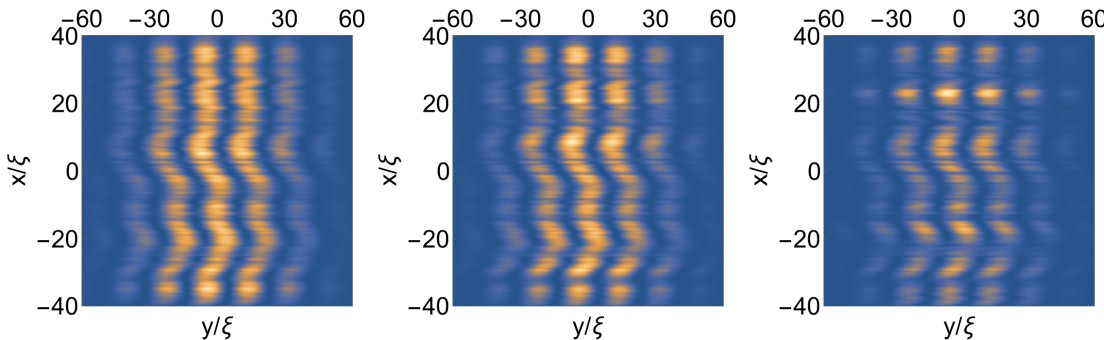

Figure 6: Outcomes for projective measurements of $\hat{\rho}_{\mathrm{tof}}(x, \vec{r}, t_1 + t_0)$, with $t_1 = 16\,\mathrm{ms}$ and $t_0 = 14\,\xi/v$, created using (29). From left to right, the temperatures are $k_{\mathrm{B}}T = 0.1\,\hbar\omega_\perp$, $k_{\mathrm{B}}T = 0.3\,\hbar\omega_\perp$ and $k_{\mathrm{B}}T = 0.7\,\hbar\omega_\perp$. To allow for an easy comparison, the same eigenvalue $\varphi_a(x)$ has been used throughout, whereas the eigenvalues $\varphi_s(x)$ are drawn from shot to shot, using the temperature-dependent distribution functions for the symmetric sector computed in Section 5.1. The other parameters used here are as presented in Section 5.3.

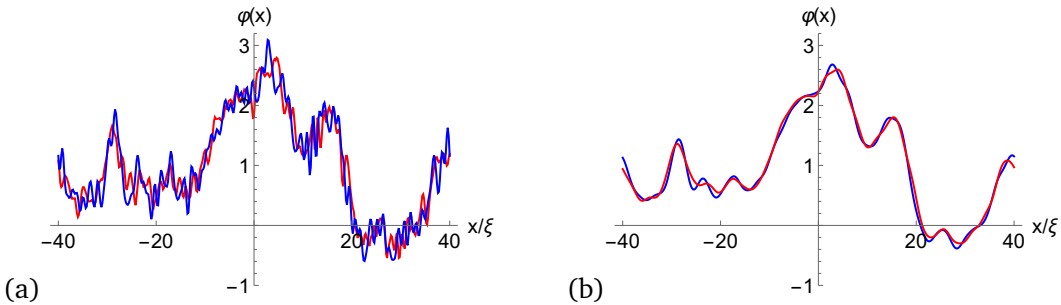

Figure 7: (a) Individual realization of the eigenvalue $\varphi_a(x)$ (*blue*), compared to the extracted phase $\tilde{\varphi}_a(x)$ (*red*) at time of flight $t_1 = 4\,\mathrm{ms}$ and $t_0 = 14\,\xi/v$. (b) The same objects, convolved with a Gaussian kernel of width $\xi = \hbar\pi/mv$. The parameters used here are presented in Section 5.3, with $k_{\mathrm{B}}T = 0.5\,\hbar\omega_\perp$, so that $T \approx 34\,\mathrm{nK}$.

presence of $e^{i\varphi_s(x,t_0)}$ and the double convolution with a Green's function. Such complications do not exist for the simplified fit formula (50), which neglects longitudinal expansion. This raises the question how good the results are if, after measuring a density profile given by (30), one still uses eqn (50) to extract an approximate eigenvalue $e^{i\tilde{\varphi}_a(x,t_0)}$. Having both the full and approximate expressions at hand, we can explicitly investigate the accuracy of such an analysis. This is of considerable importance for the analysis of experiments. To this end, we draw an eigenvalue $e^{i\varphi_a(x,t_0)}$ from the distribution function computed in Section (5.1), and construct the corresponding density profile using (30). We then use the simplified fit formula (50) to extract an approximate eigenvalue $e^{i\tilde{\varphi}_a(x,t_0)}$. This can then be compared to the original, exact eigenvalue $e^{i\varphi_a(x,t_0)}$. Figs 7, 8, 9, 10 show representative examples of such comparisons.

In Fig. 7(a) the extracted phase $\tilde{\varphi}_a(x)$ (red) is compared to the exact phase $\varphi_a(x)$ (blue). Although the results clearly deviate, most of these deviations occur on small lengthscales, which are not observed in experiment. To remove these short wave length fluctuations we convolve the signal with a Gaussian kernel of width $\xi$. The resulting smoothened curves are seen to be in good agreement for short time of flight (Fig. 7, with $t_1 = 4\,\mathrm{ms}$), whereas significant deviations do occur for long flight times (Fig. 10, with $t_1 = 32\,\mathrm{ms}$). The size of these deviations does not depend strongly on the temperature, which only enters through the fields in the symmetric sector. These symmetric sector fields have an effect on the amplitude

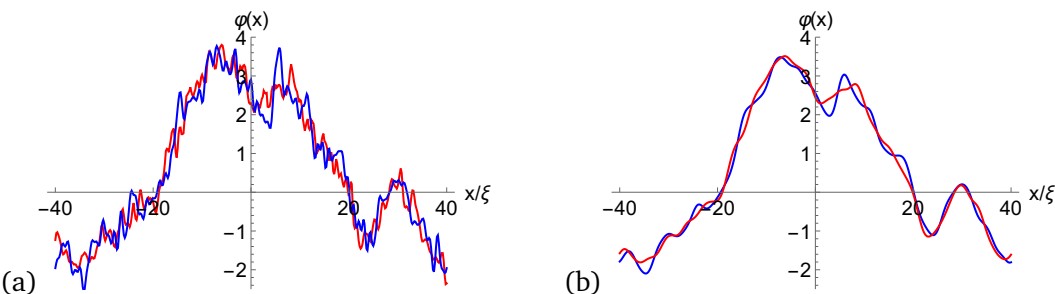

Figure 8: The same as Fig. 7, but at time of flight $t_1 = 8\,\text{ms}$, and for a different phase eigenvalue.

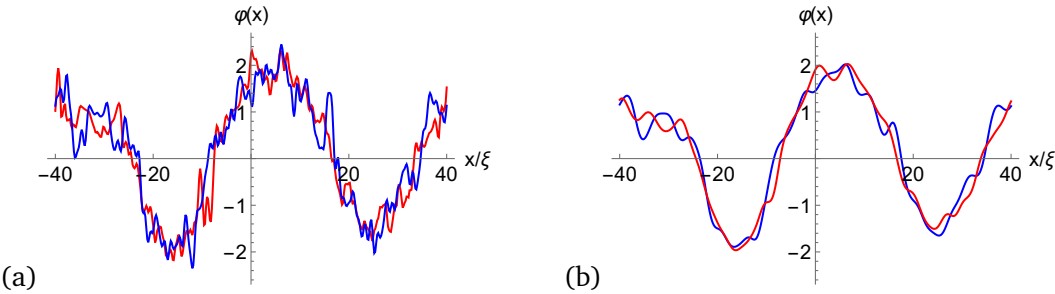

Figure 9: The same as Fig. 7, but at time of flight $t_1 = 16\,\text{ms}$, and for a different phase eigenvalue.

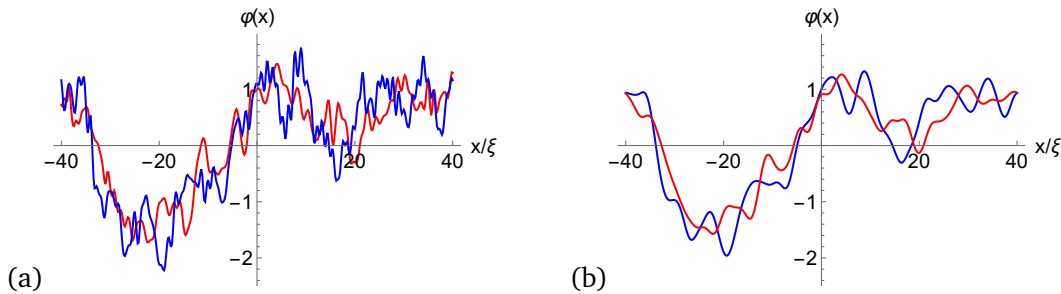

Figure 10: The same as Fig. 7, but at time of flight $t_1 = 32\,\text{ms}$, and for a different phase eigenvalue.

of the density ripples, but not on the transverse position of the fringes, as can be understood by inspection of eqn (30): the eigenvalue $e^{i\varphi_s(x)}$ appears in both terms in parentheses, so that it does not affect the interference term independently. For this reason, spatial fluctuations in the eigenvalue $e^{i\varphi_s(x)}$ do not strongly impede the reconstruction of the eigenvalue $e^{i\varphi_a(x)}$.

The above analysis leads us to conclude that at sufficiently short times of flight the simplified fit formula (50) can be used to obtain an accurate approximation to the eigenvalues $e^{i\varphi_a(x,t_0)}$.

In order to compare to experimental data one also should model the effects of the trapping potential. This can be done in the framework of a local density approximation [44, 46–49]. We refrain from presenting such an analysis here, but instead simply introduce an overall suppression $e^{-x^2/(L/4)^2}$ along the length of the gas. In Fig. 11 we present a comparison of theoretical results obtained in this way to experimental data from Ref. [38]. We see that the theoretical result reproduces the various structures seen in experiment. Due to the statistical nature of measurements in quantum theory the outcome shown in the theoretical plot is of course not expected to coincide with that of the experimental plot.

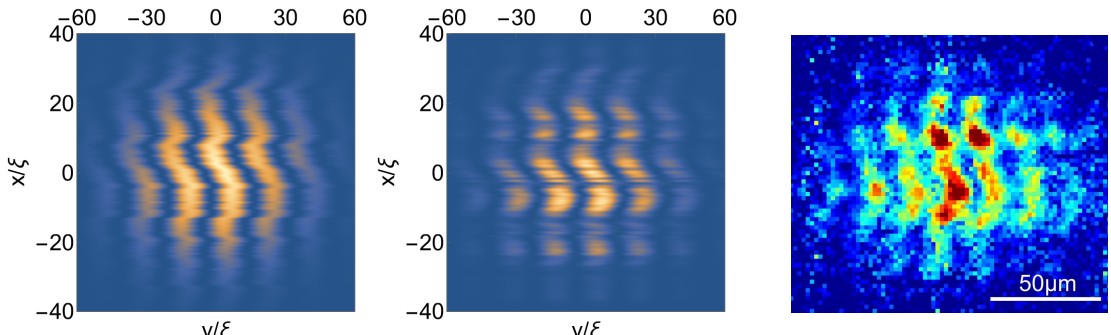

Figure 11: (Left) Theoretical results for individual measurement outcomes $\varrho_{\text{tof}}(\vec{r}, t_1)$ at $t_0 = 14\,\xi/v$ and $t_1 = 16\,\text{ms}$. An overall suppression with a factor $e^{-x^2/(L/4)^2}$ has been applied along the length of the gas (see main text). Longitudinal expansion during time-of-flight has been neglected and the parameters are described in Section 5.3, with $k_B T = 0.5\,\hbar\omega_\perp$, so that $T \approx 34\,\text{nK}$. (Middle) Same as left panel but with longitudinal expansion taken into account using (29). (Right) Experimentally measured density profile taken from Ref. [38].

## 7  Conclusions

In this work we have revisited the theoretical description of the measurement process involved in time-of-flight recombination of split one-dimensional Bose gases. We have derived the relation between the measured density operator after expansion and local operators in the Luttinger liquid theory describing the low energy degrees of freedom in such systems. In the weakly interacting regime and in cases where the longitudinal expansion can be neglected the measured density is related in a simple way to a vertex operator of the phase field in a Luttinger liquid. We have discussed the theoretical description of individual (projective) measurements in this setting. To the best of our knowledge this issue has not been previously addressed in the literature. We also have described how multi-point correlation functions of vertex operators can be extracted from projective measurements of the boson density in time of flight experiments. Our main new result, which is of direct relevance for experiments, is the description of projective density measurements in the framework of Luttinger liquid theory in the case of weak interactions but non-negligible longitudinal expansion of the gas after the trap release. Here the main new effect is that phase fluctuations in the symmetric sector induce intensity variations along the interference fringes ("density ripples"), the magnitude of which increases with time of flight. As an explicit example we considered the case of weakly interacting coherently split Bose gases in the absence of tunnel coupling. In this case the time evolution can be analyzed explicitly in the framework of Luttinger liquid theory, see e.g. [36]. Our results for a single measurement reproduce all the main features seen in experiment. The theoretical framework developed here applies equally to the case of weakly interacting split condensates in the presence of a weak tunnel coupling. Here the antisymmetric sector of the theory is described by a quantum sine-Gordon model in the weak interaction regime and the time evolution can no longer be analyzed in a simple fashion. Our work raises a number of interesting questions. First and foremost our result (30) suggests that it should be possible to extract information on the symmetric sector of the theory from the density ripples along the interference fringes. An investigation of this issue is under way. Having direct experimental access to properties of the symmetric sector is important as the existing theoretical analyses suggest that the relaxational behaviour of the symmetric sector is very different from that of the antisymmetric sector.

## Acknowledgements

We are grateful to the Erwin Schrödinger International Institute for Mathematics and Physics for hospitality and support during the programme on *Quantum Paths*. This work was supported by the EPSRC under grant EP/N01930X (FHLE) and YDvN is supported by the Merton College Buckee Scholarship and the VSB and Muller Foundations. JS acknowledges support by the European Research Council, ERC-AdG *QuantumRelax* (320975).

## A    Relation between density operators before and after release

We here present the details of the derivation of eqn (7), by performing the integrals in (5),

$$\hat{\Psi}_{\text{tof}}(x,\vec{r},t_1+t_0) = \int \frac{dk\,d^2\vec{p}\,dy\,d^2\tilde{\vec{r}}}{(2\pi)^3} e^{-ik(x-y)} e^{-i\vec{p}\cdot(\vec{r}-\tilde{\vec{r}})} e^{-it_1\frac{k^2+\vec{p}^2}{2m}} \hat{\Psi}(z,\tilde{\vec{r}},t_0), \qquad (52)$$

after insertion of relation (6),

$$\hat{\Psi}(x,\vec{r},t_0) = \hat{\psi}_1(x,t_0)g(\vec{r}+\vec{d}/2) + \hat{\psi}_2(x,t_0)g(\vec{r}-\vec{d}/2), \qquad (53)$$

where $g_1(\vec{r})$ is the ground state wave function of a two-dimensional harmonic oscillator with frequency $\omega$,

$$g_1(\vec{r}) = \sqrt{\frac{m\omega}{\pi}} e^{-\frac{m\omega}{2}\vec{r}^2}. \qquad (54)$$

Defining $\psi_1 \equiv \psi_-$ and $\psi_2 \equiv \psi_+$ and carrying out the integrals we have

$$\hat{\Psi}_{\text{tof}}(x,\vec{r},t_1+t_0) = \sum_{\pm} \sqrt{\frac{m\omega}{\pi}} \frac{e^{-i\pi/2} e^{i\arctan\frac{1}{\omega t_1}}}{\sqrt{1+\omega^2 t_1^2}} \exp\left(-\frac{m\omega}{2}\frac{\left(\vec{r}\pm\vec{d}/2\right)^2}{1+\omega^2 t_1^2}\right) \times$$
$$\times \exp\left(i\frac{m\omega^2 t_1}{2\left(1+\omega^2 t_1^2\right)}\left(\vec{r}\pm\vec{d}/2\right)^2\right) \int dy\, G\left(x-y,t_1\right)\hat{\psi}_{\pm}(y,t_0), \qquad (55)$$

where we have defined the free, single-particle Green's function

$$G(y,t) = \int \frac{dk}{2\pi} e^{-iky} e^{-i\frac{t\gamma}{2m}k^2} = \begin{cases} \sqrt{\frac{m}{2\pi i t\gamma}} \exp\left(i\frac{m}{2t\gamma}y^2\right), & \text{if } \gamma = 1 \\ \delta(y), & \text{if } \gamma = 0. \end{cases} \qquad (56)$$

We are interested in the limit of a very narrow trapping potential. Assuming that $\omega t_1 \gg 1$ and $|\vec{r}| \gg |\vec{d}|$ we may simplify (55) further, to

$$\hat{\Psi}_{\text{tof}}(x,\vec{r},t_1+t_0) \approx -i \sum_{\pm} \hat{\psi}_{\pm}(x) \sqrt{\frac{m\omega}{\pi(1+\omega^2 t_1^2)}} \exp\left(-\frac{m\omega}{2}\frac{\vec{r}^2}{1+\omega^2 t_1^2}\right) \times$$
$$\times \exp\left(i\frac{m}{2t_1}\left(\vec{r}\pm\vec{d}/2\right)^2\right) \int dy\, G\left(x-y,t_1\right)\hat{\psi}_{\pm}(y,t_0). \qquad (57)$$

From this expression, we recover eqn (7) with

$$f(\vec{r},t_1) = -i\sqrt{\frac{m\omega}{\pi}} \frac{1}{\sqrt{1+\omega^2 t_1^2}} \exp\left(-\frac{m\omega}{2}\frac{\vec{r}^2}{1+\omega^2 t_1^2}\right). \qquad (58)$$

# B  Bosonization conventions

The low-energy physics of the microscopic Hamiltonian (13) is described by a *Luttinger liquid* [36, 41, 50] with Hamiltonian

$$H_{\text{LL}} = \frac{v}{2\pi} \sum_{j=s,a} \int_{-L/2}^{L/2} dx \left[ K(\partial_x \hat{\phi}_j(x))^2 + \frac{1}{K} \left( \partial_x \hat{\theta}_j(x) \right)^2 \right]. \tag{59}$$

The (real) fields $\hat{\phi}_{a,s}$ and $\hat{\theta}_{a,s}$ are related to the original complex bosons $\psi_{1,2}$ by the transformation (17) and the bosonization identity

$$\psi_j^\dagger(x) \sim \sqrt{\rho_0 + \frac{\partial_x \hat{\theta}_j(x)}{\pi}} \, e^{-i\hat{\phi}_j(x)} \sum_m A_m e^{2im\left(\hat{\theta}_j(x) + \pi\rho_0 x\right)}, \quad j = 1, 2. \tag{60}$$

Here $A_m$ are non-universal coefficients, $\partial_x \hat{\theta}_{1,2}$ describe density fluctuations and $\hat{\phi}_{1,2}$ are phase fields. They satisfy canonical commutation relations

$$\left[ \frac{\partial_x \hat{\theta}_i(x)}{\pi}, \hat{\phi}_j(z) \right] = i\delta_{i,j}\delta(x-z). \tag{61}$$

The cutoff length scale for the low-energy field theory (59) is set by the healing length of the gas, which for weak interactions reads $\xi = \pi/mv$. The Hamiltonian (59) is parametrized by the velocity $v$ and the *Luttinger parameter*, $K$. For weak interactions they are related to the parameters of the microscopic Hamiltonian (13) as follows [50]

$$v = \frac{\rho_0}{m} \sqrt{\gamma} \left( 1 - \frac{\sqrt{\gamma}}{2\pi} \right)^{1/2}, \quad K = \frac{\pi}{2\sqrt{\gamma}} \left( 1 - \frac{\sqrt{\gamma}}{2\pi} \right)^{-1/2}, \quad \rho_0 = \frac{2mvK}{\pi}, \tag{62}$$

where we have used the dimensionless parameter $\gamma = mg/\rho_0$.

We use periodic boundary conditions throughout this paper. As $\hat{\phi}_{a,s}$ are compact fields we have

$$\hat{\phi}_{a,s}(x + L) = \hat{\phi}_{a,s}(x) + 2\pi \hat{J}_{a,s}, \tag{63}$$

where the eigenvalues of $\hat{J}_{a,s}$ are integers related to the number of times the phase winds around a circle of radius $2\pi$ over the length of the gas. The density operator has to satisfy

$$\int_0^L dx \, \partial_x \hat{\theta}_{a,s} = \pi \delta \hat{N}_{a,s}, \tag{64}$$

where $\delta \hat{N}_{a,s}$ has integer eigenvalues which count the particle imbalance in the symmetric and antisymmetric sectors respectively. These considerations lead to the mode expansions

$$\hat{\theta}_j(x) = \hat{\theta}_{j,0} + \frac{\pi x}{L} \delta \hat{N}_j + \sum_{q \neq 0} \left| \frac{\pi K}{2qL} \right|^{1/2} e^{iqx} \left( \hat{a}_{j,q} + \hat{a}_{j,-q}^\dagger \right), \tag{65}$$

$$\hat{\phi}_j(x) = \hat{\phi}_{j,0} + \frac{\pi x}{L} \hat{J}_j + \sum_{q \neq 0} \left| \frac{\pi}{2qLK} \right|^{1/2} \text{sgn}(q) e^{iqx} \left( \hat{a}_{j,q} - \hat{a}_{j,-q}^\dagger \right), \tag{66}$$

where $\hat{a}_{i,q}$ are oscillator modes with commutation relations $[\hat{a}_{i,q}, \hat{a}_{j,k}^\dagger] = \delta_{q,k}\delta_{i,j}$, and $[\delta \hat{N}, \hat{\phi}_0] = i = [\hat{J}, \hat{\theta}_0]$. The momenta are quantized as $q_n = 2\pi n/L$. The mode expansion of the Hamiltonian (59) is

$$H_{\text{LL}} = \sum_{j=a,s} \left[ \frac{\pi v K \hat{J}_j^2}{2L} + \frac{\pi v (\delta \hat{N}_j)^2}{2KL} + \sum_{q \neq 0} v|q| \hat{a}_{j,q}^\dagger \hat{a}_{j,q} \right]. \tag{67}$$

For our purposes it will suffice to consider only the $\hat{J} = 0$ subspace. The rationale for this is that $\hat{J}$ has eigenvalue zero for all experimentally relevant initial states and the Hamiltonians we consider commute with $\hat{J}$.

A compact notation for the zero modes used in eqns (31-32) is to introduce annihilation operators

$$\hat{a}_{a,0} = -i\sqrt{\frac{2K}{v}}\hat{\phi}_{a,0} - \frac{1}{2}\sqrt{\frac{v}{2K}}\delta\hat{N}_a \, . \tag{68}$$

## C   Normalization of vertex operator eigenstates

We here derive eqns (37) and (38). In order to regulate the infinity caused by the delta function, we consider the following modification of the state (36)

$$|\{f_n\}\rangle_\tau = \mathcal{N}_f \exp\sum_k\left(\frac{\tau}{2}\hat{a}_k^\dagger\hat{a}_{-k}^\dagger + \frac{f_k}{u_k}\hat{a}_k^\dagger\right)|0\rangle \, , \tag{69}$$

and recover the eventual delta function normalization by taking the limit $\tau \to 1$ at the end of the calculation. Our task is to calculate the overlap

$$
\begin{aligned}
\langle g|f\rangle &= {}_\tau\langle\{g_n\}|\{f_n\}\rangle_\tau \\
&= \mathcal{N}_g^*\mathcal{N}_f\langle 0|\exp\left(\sum_j\frac{\tau}{2}\hat{a}_j\hat{a}_{-j} + \frac{g_j^*}{u_j^*}\hat{a}_j\right)\exp\left(\sum_k\frac{\tau}{2}\hat{a}_k^\dagger\hat{a}_{-k}^\dagger + \frac{f_k}{u_k}\hat{a}_k^\dagger\right)|0\rangle \, .
\end{aligned} \tag{70}
$$

Inserting a resolution of the identity in terms of coherent states

$$|\alpha\rangle = \prod_k e^{-|\alpha_k|^2/2}e^{\alpha_k\hat{a}_k^\dagger}|0\rangle \, , \qquad \mathbb{1} = \int D(\alpha,\alpha^*)|\alpha\rangle\langle\alpha| \tag{71}$$

with $D(\alpha,\alpha^*) = \prod_k d\mathrm{Re}\alpha_k\, d\mathrm{Im}\alpha_k/\pi$ and using that $\hat{a}_k|\alpha\rangle = \alpha_k|\alpha\rangle$, we have

$$\langle g|f\rangle = \int D(\alpha,\alpha^*)\mathcal{N}_g^*\mathcal{N}_f\exp\sum_j\left(-\alpha_j\alpha_j^* + \frac{\tau}{2}\alpha_j\alpha_{-j} + \frac{g_j^*}{u_j^*}\alpha_j + \frac{\tau}{2}\alpha_j^*\alpha_{-j}^* + \frac{f_j}{u_j}\alpha_k^*\right) \, . \tag{72}$$

Noting that $u_j$ satisfies

$$\begin{cases} \mathrm{Im}(u_j) = 0, & u_{-j} = -u_j, & \text{if } j \neq 0, \\ \mathrm{Re}(u_0) = 0, & & \text{else,} \end{cases} \tag{73}$$

and using $f_{-n}^* = f_n$ and $f_0^* = f_0$ we can carry out the integrals. Finally we use that

$$\lim_{\epsilon\to 0}\frac{1}{(2\pi\epsilon)^{d/2}}e^{-\frac{|x|^2}{2\epsilon}} = \delta^{(d)}(|x|) \tag{74}$$

to arrive at

$$
\begin{aligned}
\lim_{\tau\to 1}\langle\{g_n\}|\{f_n\}\rangle_\tau &= \mathcal{N}_g^*\mathcal{N}_f\sqrt{2\pi}|u_0|\exp\left(\frac{1}{8|u_0|^2}(g_0 + f_0)^2\right)\delta(g_0 - f_0)\times \\
&\quad\times\prod_{k>0}\pi|u_k|^2\exp\left(\frac{1}{4|u_k|^2}|g_k + f_k|^2\right)\delta^{(2)}(g_k - f_k) \, .
\end{aligned} \tag{75}
$$

This shows that the states $|\{f_n\}\rangle$ are delta-normalized if the normalization constants $\mathcal{N}_f$ are chosen according to eqn (37).

## D  Time-dependent overlap for the zero mode

The zero mode initial state $|\psi_{k=0}\rangle$ is determined by the overlap

$$\langle n|\psi_{k=0}\rangle = \left(\frac{1}{\pi\rho_0 L}\right)^{1/4} \exp\left(-\frac{1}{2\rho_0 L}n^2\right), \tag{76}$$

where $|n\rangle$ is the eigenstate of $\delta\hat{N}$ with eigenvalue $n$. The operators $\delta\hat{N}$ and $\hat{\phi}_0$ satisfy canonical commutation relations, $\left[\delta\hat{N},\hat{\phi}_0\right]=i$. In analogy with eigenstates of the $\hat{x}$- and $\hat{p}$-operators in quantum mechanics, this means that the eigenstate $|f_0\rangle$ of $\hat{\phi}_0$ has an overlap with the eigenstate $|n\rangle$ of $\delta\hat{N}$ which is given by

$$\langle n|f_0\rangle = \frac{e^{inf_0}}{\sqrt{2\pi}}. \tag{77}$$

We are interested in computing the time-dependent overlap

$$\langle f_0|\psi_{k=0}(t)\rangle = \langle f_0| e^{-iH_{k=0}t} |\psi_{k=0}\rangle. \tag{78}$$

Since the zero mode part of the Hamiltonian is given by

$$H_{k=0} = \frac{\pi v(\delta\hat{N})^2}{2KL}, \tag{79}$$

its action on the state $|n\rangle$ is trivial, and we can compute the time-dependent overlap by inserting a complete set of such states. This leads to the result that

$$\langle f_0| e^{-iH_{k=0}t} |\psi_{k=0}\rangle = \int dn \, \langle f_0|n\rangle \, \langle n| e^{-iH_{k=0}t} |\psi_{k=0}\rangle$$

$$= \left(\frac{1}{\pi\rho_0 L}\right)^{1/4} \frac{1}{\sqrt{\frac{1}{\rho_0 L} + i\frac{\pi v t}{KL}}} \exp\left(-\frac{1}{2}\frac{f_0^2}{\frac{1}{\rho_0 L} + i\frac{\pi v t}{KL}}\right). \tag{80}$$

## E  Overlap with a general Fock state

We here compute the overlaps between a generic phase eigenstate (36) and a Fock state $|\{n_{q\neq 0}\}\rangle$, where we assume that the occupation numbers satisfy $n_q = n_{-q}$. The zero mode will not be treated here. Defining

$$\mathcal{N}_q = \left(\frac{1}{\pi|u_q|^2}\right)^{1/2} e^{-\frac{1}{2|u_q|^2}|f_q|^2}, \tag{81}$$

we consider sectors $(q,-q)$ separately. This leads to

$$\langle n_{-q},n_q|f_{-q},f_q\rangle = \mathcal{N}_q \, \langle n_{-q},n_q| \sum_{n=0}^{\infty} \frac{1}{n!} \left(\hat{a}_q^{\dagger}\hat{a}_{-q}^{\dagger} + \frac{f_q}{u_q}\hat{a}_q^{\dagger} + \frac{f_q^*}{u_q^*}\hat{a}_{-q}^{\dagger}\right)^n |0\rangle$$

$$= \mathcal{N}_q n_q! \sum_{n=0}^{\infty} \frac{1}{n!} \left(\frac{f_q}{u_q}\right)^{\alpha} \left(\frac{f_q^*}{u_q^*}\right)^{\gamma} C(\alpha,\gamma), \tag{82}$$

with

$$\alpha = n - n_q = \gamma, \tag{83}$$

and $n_q \leq n \leq 2n_q$. The combinatoric factors read

$$C(\alpha, \gamma) = \binom{n}{\alpha + \gamma}\binom{\alpha + \gamma}{\gamma} = \frac{n!}{(2n_q - n)!((n - n_q)!)^2}. \tag{84}$$

The overlap in the $(q, -q)$-sector is then given by

$$\langle n_{-q}, n_q | f_{-q}, f_q \rangle = \mathcal{N}_q \sum_{n=n_q}^{2n_q} \frac{n_q!}{(2n_q - n)! \left((n - n_q)!\right)^2} (-1)^{n-n_q} \left| \frac{f_q}{u_q} \right|^{2n-2n_q} = \mathcal{N}_q \, L_{n_q}\left( \left| \frac{f_q}{u_q} \right|^2 \right), \tag{85}$$

where $L_n(x)$ is the Laguerre polynomial of degree $n$. Inserting the definition of $\mathcal{N}_q$, we find the squared overlap coefficients per $(q, -q)$-sector,

$$\left| \langle n_{-q}, n_q | f_{-q}, f_q \rangle \right|^2 = \frac{1}{\pi |u_q|^2} L_{n_q}^2\left( \left| \frac{f_q}{u_q} \right|^2 \right) e^{-\left| \frac{f_q}{u_q} \right|^2}. \tag{86}$$

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
