# Peer review of "Projective phase measurements in one-dimensional Bose gases"

_SciPost Physics, doi:SciPost Phys. 5, 046 (2018)_

## Round 1 · Referee Report · Anonymous · 2018-7-18

Strengths

1 - very well written

2 - pedagogical

3 - practical

Weaknesses

1 - original results not clearly stated

Report

The paper describes interference of two quasicondensates after their release from parallel one-dimensional traps assuming no interaction during the time of flight and no trapping in the longitudinal direction. The authors relate the density profile after the expansion with the initial state of the phase and density fluctuations. They argue that such an interference experiment is a tool for extracting multi-point correlation functions. Effects of the longitudinal dynamics, which complicate this extraction, are analyzed in detail.

The paper is easy to read. The authors present their derivation step-by-step, clearly explaining their motivation, methods, and underlying assumptions. I recommend publication after minor revision. My comments and suggestions are listed below.

Requested changes

1) My main concern is that although the paper reads smoothly, it is difficult to tell new results from the state of the art. Can the authors be more specific about their original contribution? The abstract is particularly not informative in this respect.

2) How good is the assumption that the atoms are noninteracting after the release? Is there a quantitative condition for this? I think that for excitations at a sufficiently high momentum the drop of the mean-field interaction may seem almost adiabatic.

3) Choose either $\mathfrak{R}$ or $\rm Re$ to denote the real part. Right now both are used (see, for example, Eqs.(45) and (49)).

  • validity: top
  • significance: high
  • originality: ok
  • clarity: top
  • formatting: excellent
  • grammar: perfect

Author:  Yuri Daniel van Nieuwkerk  on 2018-09-03  [id 314]

(in reply to Report 1 on 2018-07-18)

We thank the referees for their helpful comments. We have addressed the various points raised by the referees as follows:

1) We have changed the abstract and conclusion section in order to make clear what our original contributions are. Our main new results are (i) an expression for the measured particle density after trap release in terms of convolutions of the eigenvalues of vertex operators involving both sectors of the two-component Luttinger liquid that describes the low-energy regime of the split condensate; and (ii) obtaining and presenting results for single-shot projective measurements.

2) The question why interactions after release can be neglected is addressed in Imambekov et al. (2009), to which we have added a reference in the appropriate section.

3) We have uniformized our notations for real parts.

---

## Round 1 · Referee Report · Karen Kheruntsyan · 2018-8-13

Strengths

1. Pedagogical
2. Clearly written
3. Clarifies aspects of previous works in a stand-alone article, which will be useful to nonspecialists

Weaknesses

1. No major weaknesses, except for some minor issues with definitions.

Report

The authors have revisited the theoretical description of observables involved in time-of-flight measurements of particle density of coherently split one-dimensional Bose gases. They provide a nice piece of pedagogical account of how such measurements relate to the eigenvalues of the vertex operators of the phase filed and how they can be used to extract multi-point correlation functions of vertex operators. The authors also clearly elaborate on the theoretical assumptions underlying the analysis of interference patterns emerging in these experiments. The paper is clearly written and in my opinion warrants publication in SciPost. I feel, however, that it can be further strengthened if the authors address the following comments and suggestions:

Requested changes

1. The authors use units in which $\hbar=1$ right from the start of the article. It would be nice if this was clearly stated.

2. After Eq. (15), the authors express the healing length $\xi$ in terms of $v$ which is not defined. The speed (presumably of sound) v is defined later in the appendix, but it would be good if it was also defined (at least spelled out) in the main text as well.

3. Similar suggestion applies to the Luttinger liquid parameter K in Eq. (16), and the tunnel-coupling strengths $\lambda$ and $\lambda'$ in Eqs. (18) and (19), which are not defined.

4. The authors carefully examine the cases when the longitudinal expansion can be neglected or when it is included in the analysis. The term "longitudinal expansion" is used throughout the papers, however, the authors never spell out what such an expansion physically actually amounts to in their model, given that the original Hamiltonian, Eq. (13), is formulated as a uniform 1D system of FIXED length L with periodic boundary condition. Given that in real experiments, "longitudinal expansion" refers to an actual physical expansion of a finite-length inhomogeneous gas (initially trapped longitudinally as well, by a harmonic trap), it would be good to clarify for nonspecialist readers what "longitudinal expansion" (phase dynamics, current flow along $x$?) refers to in here, and how this can be incorporated within the framework of the local density approximation which the authors mention towards the end of the paper.

  • validity: high
  • significance: high
  • originality: ok
  • clarity: high
  • formatting: excellent
  • grammar: excellent

Author:  Yuri Daniel van Nieuwkerk  on 2018-09-03  [id 315]

(in reply to Report 2 by Karen Kheruntsyan on 2018-08-13)

We thank the referees for their helpful comments. We have addressed the various points raised by the referees as follows:

(1) We have added a statement that we use units in which $\hbar=1$.

(2) and (3) We have clarified our definitions of the speed of sound $v$, the Luttinger parameter $K$ and the tunnelling strength $\lambda$.

(4) We have clarified what we mean by “longitudinal expansion” and added a comment explaining why this remains a meaningful concept even though we use periodic boundary conditions for simplicity.

---

## Round 2 · Referee Report · Anonymous (Referee 1) · 2018-9-26

Report

I am satisfied with the author's response to my criticism and I recommend the paper for publication.

---

## Round 2 · Referee Report · Anonymous (Referee 3) · 2018-10-3

Report

I am satisfied with the author's responses and recommend the paper for publication.

---

## Round 2 · Author Response

We thank the referees for their helpful comments. We have addressed the various points raised by the referees as follows:

Referee 1:

(1) We have changed the abstract and conclusion section in order to make clear what our original contributions are. Our main new results are (i) an expression for the measured particle density after trap release in terms of convolutions of the eigenvalues of vertex operators involving both sectors of the two-component Luttinger liquid that describes the low-energy regime of the split condensate; and (ii) obtaining and presenting results for single-shot projective measurements.

(2) The question why interactions after release can be neglected is addressed in Imambekov et al. (2009), to which we have added a reference in the appropriate section.

(3) We have uniformized our notations for real parts.

Referee 2:

(1) We have added a statement that we use units in which $\hbar=1$.

(2) and (3) We have clarified our definitions of the speed of sound $v$, the Luttinger parameter $K$ and the tunnelling strength $\lambda$.

(4) We have clarified what we mean by “longitudinal expansion” and added a comment explaining why this remains a meaningful concept even though we use periodic boundary conditions for simplicity.

---

## Round 2 · List of Changes

-We have rewritten the Abstract to more clearly reflect our original contributions. We have made similar changes to the Introduction and Conclusion.
-We have added a remark about the use of units in which $\hbar=1$. Also, we have clarified our definitions of the speed of sound $v$, the Luttinger parameter $K$ and the tunnelling strength $\lambda$. The notation for real parts of complex numbers has been uniformized.
-The question why interactions after release can be neglected is addressed in Imambekov et al. (2009), to which we have added a reference in the appropriate section.
-We have rewritten the first two paragraphs of Section 2 to clarify what we mean by “longitudinal expansion” and added a comment explaining why this remains a meaningful concept even though we use periodic boundary conditions for simplicity. A related comment has been added at the beginning of Section 3.
-We have added remarks to the Abstract and Conclusion about the possibility of extracting information about the symmetric sector from the interference pattern, as a perspective for further research.

---

## Editorial Decision

published